# Evaluating Feasibility of Human Tissue Engineered Respiratory Epithelium Construct as a Potential Model for Tracheal Mucosal Reconstruction

**DOI:** 10.3390/molecules26216724

**Published:** 2021-11-06

**Authors:** Mohd Heikal Mohd Yunus, Zahra Rashidbenam, Mh Busra Fauzi, Ruszymah Bt Hj Idrus, Aminuddin Bin Saim

**Affiliations:** 1Department of Physiology, Faculty of Medicine, UKM Medical Centre, Jalan Yaacob Latiff, Cheras, Kuala Lumpur 56000, Malaysia; ruszyidrus@gmail.com; 2Centre for Tissue Engineering and Regenerative Medicine, Faculty of Medicine, UKM Medical Centre, Jalan Yaacob Latiff, Bandar Tun Razak, Kuala Lumpur 56000, Malaysia; Zahra.rashidbenam@gmail.com (Z.R.); fauzibusra@ukm.edu.my (M.B.F.); 3Ear, Nose & Throat Consultation Clinic, Ampang Puteri Specialist Hospital, Ampang 68000, Selangor, Malaysia; aminuddinsaim@gmail.com

**Keywords:** respiratory epithelium, polymerized-human plasma, respiratory epithelial construct tracheal mucosal reconstruction

## Abstract

The normal function of the airway epithelium is vital for the host’s well-being. Conditions that might compromise the structure and functionality of the airway epithelium include congenital tracheal anomalies, infection, trauma and post-intubation injuries. Recently, the onset of COVID-19 and its complications in managing respiratory failure further intensified the need for tracheal tissue replacement. Thus far, plenty of naturally derived, synthetic or allogeneic materials have been studied for their applicability in tracheal tissue replacement. However, a reliable tracheal replacement material is missing. Therefore, this study used a tissue engineering approach for constructing tracheal tissue. Human respiratory epithelial cells (RECs) were isolated from nasal turbinate, and the cells were incorporated into a calcium chloride-polymerized human blood plasma to form a human tissue respiratory epithelial construct (HTREC). The quality of HTREC in vitro, focusing on the cellular proliferation, differentiation and distribution of the RECs, was examined using histological, gene expression and immunocytochemical analysis. Histological analysis showed a homogenous distribution of RECs within the HTREC, with increased proliferation of the residing RECs within 4 days of investigation. Gene expression analysis revealed a significant increase (*p* < 0.05) in gene expression level of proliferative and respiratory epithelial-specific markers Ki67 and MUC5B, respectively, within 4 days of investigation. Immunohistochemical analysis also confirmed the expression of Ki67 and MUC5AC markers in residing RECs within the HTREC. The findings show that calcium chloride-polymerized human blood plasma is a suitable material, which supports viability, proliferation and mucin secreting phenotype of RECs, and this suggests that HTREC can be a potential candidate for respiratory epithelial tissue reconstruction.

## 1. Introduction

The airway epithelium serves as an interaction surface between the organism and a harsh environment, and its normal physiological function is vital for the host’s well-being. Diseases that involve tracheal tissues include congenital tracheal anomalies [1], tracheal neoplasm [2], infection [3], inflammation [4], trauma, and post-intubation injuries [5]. Treatments utilizing foreign materials such as titanium mesh [6,7], naturally derived materials such as cellulose [8], nonviable allograft [9], tissue engineering [10] and various types of tracheal transplantation have been tried. Application of such grafts often results in unfavorable outcomes such as chronic inflammation [7], immunogenicity [11] and non-degradability [12] in the human body. However, there is no predictable and dependable tracheal replacement method that has withstood long-term clinical use in any safe and practicable manner. Various animal models have been used to try various types of treatments [13,14], and this has extended to human use as well [15]. In a sheep study model, a tissue-engineered respiratory epithelium construct consisting of autologous respiratory epithelial cells, fibroblasts and sheep’s blood plasma has been proven to be supportive of cilia formation and can possibly be used as a replacement for tracheal mucosal defects [16,17,18]. Tissue-engineering technology promises ideal alternatives in treating these diseases that result in fatality if left untreated, owing to the function of the trachea as a vital air-conducting organ. Although organ transplantation is a popular treatment, the medical profession faces the problem of an overall shortage of available donor tissue, which results in patients dying before an appropriately matched organ can be found, in addition to the complications (i.e., infection and cardiovascular diseases and bone marrow suppression [19]) arising as a result of using immunosuppressant drugs in recipients [20].

The outbreak of the novel coronavirus disease COVID-19, originated from Wuhan, China, in December 2019 [21], further intensified the urgency to supply tracheal tissue replacement. Following rapid transmission of the disease across the globe, the World Health Organization (WHO) had declared the illness as a controllable pandemic [22]. Infection with COVID-19 resulted in developing a cluster of symptoms such as fever or even fatal respiratory illness such as respiratory distress syndrome and acute respiratory failure [23,24]. Tracheal intubation in COVID-19 patients with respiratory failure is among emergency airway management standard procedures [25,26], and one of the complications associated with tracheal intubation is tracheal rupture or destruction, which afterward requires tracheal replacement [27,28].

The first requirement of tissue engineering is to use cells with the correct phenotype in order to guide the formation of functional tissue. Difficulties in maintaining the mucocilated property of airway epithelial cells isolated from primary sources such as nasal turbinate and, on the other hand, non-reproducible and often complex differentiation procedures to yield airway epithelial cells are the major existing challenges in the field [29]. In addition to the complication of differentiating stem or progenitor cells into the correct phenotypes, the co-ordination of differentiated cells into a functional assembly of tissues is another vital prerequisite [20]. Meanwhile, many researchers are focusing their study on the development of in vitro models of the respiratory epithelium to study cell interactions and differentiation and mechanisms of protein production. In tissue engineering, understanding the cells at their molecular level is important in order to maintain the cell’s quality and functionality for regenerating a superior quality epithelium for in vivo use. The mechanical property of such tissue-engineered epithelial construct and its ability to withstand physiological pressure changes imposed by the process of respiration [30] are also equally important aspects to be considered.

In a study by Kojima and his colleagues (2003), it was proved that implanted tissue engineered respiratory epithelium construct was able to treat tracheal mucosal defect in a sheep model. In their study, a combination of chondrocytes and epithelial cells isolated from sheep nasal septum on a basement of glucolic acid were used for the fabrication of trachea [31]. In our study, we used a simple approach in the fabrication of epithelial construct by using only one cell type (respiratory epithelial cells) on the plasma basement. Both the cell isolation and the blood plasma collection are non-invasive methods, and in the future, both can be available from autologous sources, which makes the immune construct compatible with the recipient. The current study is focused on the investigation of human tissue respiratory epithelial construct (HTREC) to achieve a better understanding of the quality of such construct in vitro by means of the cellular proliferation, differentiation and distribution for the respiratory epithelial cells. This will help yield a good-quality construct for the purpose of tissue engineering of the tracheal epithelium for future treatment of tracheal defects.

To understand the quality of the HTREC, the expression of two critical genes is investigated for understanding the proliferation and differentiation of respiratory epithelial cells in the construct. The investigated genes are Ki67, a proliferation marker, and MUC5B, a marker for mucin secretion. Immunocytochemical analysis is performed using monoclonal antibodies against Ki67 and MUC5AC for the analysis of distribution and morphology of differentiated respiratory epithelial cells. MUC5B and MUC5AC were demonstrated to be the major components of human airway secretions. MUC5B are secreted primarily by the submucosal gland, as well as in the goblet cells, while MUC5AC are produced by the goblet cells [32].

## 2. Results

### 2.1. Cell Morphology

The primary culture of human RECs was successfully established at passage 1 from a nasal turbinate sample (Figure 1). The RECs maintained well-defined polygonal morphology, which is known as typical morphology of respiratory epithelial cells [17,33,34]. Furthermore, the cells were found to be actively proliferating, evidenced by their shining borders, which could be more clearly seen under higher magnification. In our previous studies, we successfully characterized the RECs isolated from nasal turbinate via gene expression (CK18 and 14, MUC5AC and Ki67) [35] and immunocytochemical analysis (acetyl β-tubulin, CK14, MUC5AC and Ki67) [35,36].

### 2.2. Histological Analysis of Human Tissue Respiratory Epithelial Construct Cell Morphology

The hematoxylin and eosin staining of the 3D human tissue respiratory epithelial construct cross-section at day 1 post-RECs incorporation (Figure 2A,B) showed that the cells were well blended with CaCl_2_-polymerised human plasma, and the distribution of the cells within the construct was homogenous. Increasing the cell number of the RECs by day 4 post-RECs incorporation (Figure 2C,D) indicates the expansion and proliferation of the residing RECs, and this further indicates the suitability of the CaCl_2_-polymerised human plasma in supporting growth and proliferation of the RECs.

### 2.3. Gene Expression Analysis

RT-PCR results on relative gene expression (Figure 3A) revealed that the expression of the Ki67 gene had been significantly (*p* < 0.05) upregulated at day 4 as compared to day 0 of incorporating the RECs into the CaCl_2_-polymerised human plasma. This shows a significant increase in proliferation of residing RECs within the HTREC and this finding is consistent with the H&E results. A similar trend was observed for the gene MUC5B (Figure 3B), in which a significant (*p* < 0.05) upregulation in gene expression levels at day 1 and day 4 as compared to day 0 had been detected in RECs residing within the HRTEC. The increase in expression of the MUC5B gene, which is associated with mucin secretion, denotes the property of CaCl_2_-polymerised human plasma in promoting the respiratory epithelial phenotype in its terminally differentiated status.

### 2.4. Immunocytochemical Analysis

Immunostaining of the HTREC with anti-Ki67 antibody (red) as the proliferation marker (Figure 4A) showed that the number of Ki67 expressing cells showed increments at day 4 as compared to day 1 post-REC incorporation to the construct. This indicates the active expansion and proliferation of the residing RECs within the HTREC, and this observation is consistent with the findings from the histology analysis (Figure 2). Immunostaining of the HTREC with anti-MUC5AC antibody (green) as the mucin secretion marker (Figure 4B) showed that the number of mucin-expressing cells increased at day 4 as compared to day 1 post-REC incorporation to the construct. This finding can be associated with either an increase in the cell number due to the proliferation of the residing RECs or the increase in the secretion of mucin by the RECs. The increase in secretion of mucin indicates a mature status of the RECs and this further shows the suitability of CaCl_2_-polymerised human plasma in supporting and maintaining the RECs in their terminally differentiated status.

### 2.5. Profile of Respiratory Epithelial Cell Population within HTREC

Results on a percentage of positive cells (Figure 5) revealed that the percentage of MUC5AC positive cells increased at day 4 (44.3% ± 4.53) as compared to day 1 (32.1% ± 3.56) of incorporating RECs within CaCl_2_-polymerised human plasma. Even though this increment in the MUC5AC positive cells percentage was not statistically significant (*p* > 0.05), it still highlights the suitability of HTREC in maintaining the mucin secretory phenotype of the residing RECs. The percentage of proliferative cells (Ki67 positive cells) was significantly increased (*p* < 0.05) over time and it was measured as 26.4% ± 3.39 and 50.3% ± 6.92 at day 1 and 4, post-incorporating RECs within CaCl_2_-polymerised human plasma, respectively. Values are presented as mean ± standard error of mean. The findings, consistent with the histology, gene expression and immunocytochemical analysis results, show the suitability of the HTREC in supporting proliferation and maintaining the mucin secretion phenotype of RECs.

## 3. Discussion

An ideal respiratory epithelium construct for clinical application should be fabricated in a manner closely resembling the native tissue [37]. In that sense, in the assessment of human tissue respiratory epithelial construct (HTREC), the respiratory epithelial cells (RECs) and the basement of the construct (CaCl_2_-polymerised human plasma) are the areas of concern. In this study, instead of tracheae as a known source for isolating respiratory epithelial cells, nasal turbinate was used. In our previous study [32], we proved that RECs from nasal turbinate could be used as a replacement to RECs isolated from the trachea. Selection of nasal turbinate over trachea was due to the following reasons: (a) nasal turbinate is harvested via non-invasive methods as compared to tracheae, which is often collected via invasive methods (i.e., tracheotomy), and this causes further stenosis and structural damage to tracheae in the tissue donor [38], (b) nasal turbinate is more readily available as compared to tracheae and (c) since nasal turbinate can be available from an autologous source, as opposed to allogeneic RECs (in which the cell donor and recipient patient are different individuals), it does not elicit an immune reaction in the tissue recipient. The RECs were isolated following an established protocol [10] by which the expression of CK14 and 18, MUC5AC and Ki67 [35] and immunocytochemical expression of markers acetyl β-tubulin, CK14, MUC5AC and Ki67 were proven [35,36]. It has been shown that knocking down CK14 results in reduced cell proliferation and delay in cell cycle progression [39]. Ki67, which is expressed in the cell nucleus in all phases of the cell cycle from the G0 phase, is a very well-known marker associated with cell proliferation [40]. Therefore, detection of CK14 and/or Ki67 expression in isolated RECs from nasal turbinate confirms the active state of cell proliferation. CK18 is the marker associated with epithelial cells [41] and is specifically expressed in respiratory tract epithelial cells. In a recent study on localizing the mucin markers expression in normal/healthy human airways, it was found that MUC5AC is specifically localized on the proximal cartilaginous airway and it is co-expressed with the club cell secretory protein [42]. Hence, detection of CK18 and MUC5AC (as a marker of mucin secretory cells) expression in isolated RECs from nasal turbinate confirms the proper and expected phenotype of isolated cells [43]. Among the expressed polymeric secreted mucin markers in the airway, the MUC5AC and MUC5B are the most abundant ones [44] and the significance of maintaining and promoting mucin secretory phenotype by RECs relies on the role they play in the first line of defense in the innate immune system. Mucin binds to infectious agents, has antioxidant, antiprotease, and antimicrobial properties [45] and the combined function of mucin and cilia clears the airway from a variety of pathogens and particles inhaled from the external environment [46].

Human blood plasma has been studied extensively for its application in tissue engineering and regenerative medicine [47]. The popularity of blood plasma applications mainly relies on its fibrinogen/fibrin contents [48]. Presence of such contents in blood plasma creates an environment that allows maintenance of normal activity of residing cells and those migrating cells from the surrounding tissues, and indeed, supporting the migration of neighboring cells to the site of tissue regeneration is one of the appealing features of blood plasma [49]. Moreover, the contents of blood plasma have angiogenic properties and can activate endothelial cells. Supplying oxygen and nutrients are necessary for cell growth and restoration of damaged tissue and, in that regard, proper angiogenesis is indeed one of the requisites for successful transplantation of any tissue-engineered construct (i.e., HTREC) [50]. The presence of fibrin in blood plasma as a protein carrier makes it possible to regulate cell responses and cell interactions within the scaffold/construct, and this mechanism is facilitated via controlled mass-dependent protein release [49,51,52]. It was reported earlier that three-dimensional fibrous scaffolds promote extracellular production, and this specifically favors cartilaginous tissue regeneration such as tracheal tissue [53]. In a study by Natarajan and his colleagues (2005), it was proven that the combination of fibrin and gelatin provides porous structures with high water absorption, and this, per se, makes the fibrin an ideal component for tissue engineering applications [54].

Plasma clot is often used for delivering stromal cells to the target site (i.e., bone defect) in clinical practice [55]. However, for tissue engineering purposes and as an alternate approach, using plasma from citrate-anticoagulated blood combined with calcium chloride as the plasma-clotting agent, is commonly practiced for the preparation of plasma gel. Calcium is a co-factor for several enzymatic steps in the coagulation process, and it is considered as a key factor for blood plasma clotting [56]. The gelled plasma (CaCl_2_-polymerised human plasma) itself holds the residing cells within and allows the migration of cells from the tissue-engineered construct to the surrounding tissues and vice versa [57]. Sadeghi-Ataabadi and his colleagues (2016) reported that even though calcium chloride (in different concentrations) does not have any significant impact on water content, tensile strength, pore size, porosity and osmolality of blood plasma, it does affect the clotting time and biodegradation rate of the scaffold in a concentration-dependent manner [58].

In our study, we utilized an established method of REC isolation from nasal turbinate, which was proven to yield cells with characteristics of the native state. Using CaCl_2_-polymerised human plasma as a scaffold provided a favorable microenvironment for RECs growth and proliferation. The scaffold could maintain and also promote the mucin secretory phenotype of residing REC for at least 4 days. Based on our gene expression data, both Ki67, as a marker of proliferation, and MUC5B, as a marker of mucin secretion, increased significantly over the period of 4 days. This indicates that both mechanisms, including increments in cell proliferation and increases in MUC5B gene expression levels, which per se causes more mucin secretion by individual RECs, contributed to increments of mucin secretion detected on day 4 of the immunohistochemical analysis. In future studies, a longer period for evaluating cell proliferation and mucin secretion by residing RECs in CaCl_2_-polymerised human plasma is necessary. Further investigations on the suitability of HTREC in supporting the cilia formation and expression of CK14 and CK18 (as markers of cell proliferation) by its residing RECs are required. The use of growth factors, such as plant sources, to enhance cell proliferation of RECs is also suggested for future explorations [59]. Moreover, since both the RECs from nasal turbinate and blood plasma can be provided from autologous sources, in which the donor and recipient are the same individuals, this eliminates the possibility of immune reaction and graft rejection in the recipient. Hence, the findings of this research state that HTREC is a promising candidate for application in respiratory epithelial reconstruction. The mechanical properties of the construct, however, require further investigations in the future.

## 4. Materials and Methods

### 4.1. Respiratory Epithelial and Fibroblast Cell Isolation and Culture

The isolation and culture of respiratory epithelial and fibroblast cells was performed as previously described [10] with slight modification. Nasal turbinate specimens discarded during turbinectomy were collected under aseptic conditions from six patients. The specimens were cleaned of mucus and blood three times using Dulbecco’s phosphate-buffered saline (DPBS, Invitrogen, Carlsbad, CA, USA) supplemented with 1% (*v*/*v*) penicillin and streptomycin (Invitrogen, Carlsbad, CA, USA). The mucosal layer was separated from the underlying bones and cut into 2 mm^3^ pieces and digested in 0.3% (*w*/*v*) collagenase type I (Worthington, Lakewood, NJ, USA) supplemented with 1% (*v*/*v*) penicillin and streptomycin (Invitrogen, Carlsbad, CA, USA) for 6 h in a shaker incubator at 37 °C. After tissue digestion, the cell suspension containing fibroblasts and respiratory epithelial cells (RECs) was centrifuged (Hettich Zentaifugen, Tuttlingen, Westphalia, Germany) for 5 min at 6500 rpm. The supernatant was discarded, and the cell pellet was resuspended in 5 to 10 mL of 0.05% Trypsin EDTA (Capricorn Scientific, Ebsdorfergrund, Germany) and incubated for 5 min at 37 °C to separate cell agglomerates into single cells. The mixture of respiratory epithelial cells and fibroblasts was cultured in defined respiratory epithelial serum-free culture medium LHC-9 (Invitrogen, Carlsbad, CA, USA), F-12 (Invitrogen, Carlsbad, CA, USA), and Dulbecco’s modified eagle’s medium (DMEM, Invitrogen, Carlsbad, CA, USA) with the 2:1:1 ratio, supplemented with 5% fetal bovine serum (FBS, Biowest, Riverside, MO, USA), (LHC-9:F-12:DMEM + 5% FBS). Cells were cultured in 2 mL medium per well in a 6-well plate and were incubated at 37 °C in a 5% CO_2_ incubator (RS Biotech, Irvine, UK) and media were changed every 2 days. Once confluent (80–90%), differential trypsinization of fibroblasts was performed using 0.05% trypsin-EDTA with 3 min incubation at 37 °C. This step allowed selective detachment of fibroblasts from the culture plate while leaving colonies of RECs in place. The REC colonies left in LHC-9 culture medium (Invitrogen, Carlsbad, CA, USA) in 6-well plates were trypsinized once they reached 80–90% confluence.

### 4.2. Human Plasma Preparation as Biomaterial

Preparation of human plasma as biomaterial for respiratory epithelial construct formation was performed as previously described [33]. A total of 50 mL of whole blood was withdrawn from 4 healthy donors (allogeneic source) via venipuncture. The whole blood then was centrifuged (Hettich Zentaifugen, Tuttlingen, Westphalia, Germany) at 5000 rpm for 5 min at 4 °C. Then, the plasma was collected and the pellet containing the blood cells and platelet was discarded. The plasma was filtered using a 0.2-μm filter unit (Sartorius, Gottingen, Germany) under aseptic conditions and was immediately stored at −20 °C prior to use.

### 4.3. Human Tissue Respiratory Epithelial Construct (HTREC) Formation

As previously described [17], approximately 2 million human RECs were incorporated into 1 mL of fresh allogeneic human plasma. This mixture was polymerized with 1 M of calcium chloride (CaCl_2_) with a concentration of 100 μL per 1 mL of plasma in a 12-well plate. Preparation of HTREC was performed with at least three biologically independent samples of human RECs and human blood plasma. The construct was maintained in the LHC-9 culture medium (Invitrogen, Carlsbad, CA, USA) until further evaluations on day 1 and 4.

### 4.4. Histological Analysis

Half of the three-dimensional (3D) HTREC incubated in culture medium was used for histological analysis. The HTREC was fixed in 10% neutral-buffered formalin in phosphate-buffered saline (PBS) (Invitrogen, Waltham, MS, USA), processed in a sample processor, embedded in paraffin and sectioned at a thickness of 5 μm using a Leica microtome (Leica Microsystem, Wetzlar, Germany). The tissue sections were rehydrated in series of decreasing concentrations of ethanol (Scharlau, Barcelona, Spain). The tissue sections were stained with hematoxylin and eosin (H&E), followed by dehydration steps in increasing concentrations of ethanol (Scharlau, Barcelona, Spain). The sections were cleared in xylene and were visualized using a light microscope (Olympus, Hamburg, Germany). Image capturing for each of the HTREC was repeated with three predetermined positions (field of view) on H&E slides (technical replicate) and a representative image was presented as a result.

### 4.5. Gene Expression Analysis

#### 4.5.1. Total RNA Extraction

To perform the gene expression analysis, the RECs from the HTREC were harvested using trypsin digestion. Total RNA of RECs was isolated using TRI Reagent^®^ (Molecular Research Centre, Cincinnati, OH, USA) and with DNase1 (Invitrogen Corporation, Carlsbad, CA, USA) application, according to the manufacturer’s recommendation. Polyacryl Carrier (Molecular Research Center, Cincinnati, OH, USA) was added in each extraction to precipitate the total RNA. The RNA pellet was then washed with 75% (*v*/*v*) ethanol and air-dried before dissolving in RNase/DNase free water (Invitrogen Corporation, Carlsbad, CA, USA). The purification of total RNA was performed using the RNeasy Kit (Qiagen, Hilden, Germany) and the purity was assessed using a spectrophotometer (Bio-Rad, Hercules, CA, USA). Samples with purity values outside the range of 1.8 to 2.0 were excluded from further analysis. The extracted RNA was stored at −80 °C immediately after extraction.

#### 4.5.2. Complementary DNA Synthesis and Real-Time Reverse Transcriptase Polymerase Chain Reaction

Master SYBR green mixture was prepared using iScript One-Step RT-PCR Kit with SYBR green (Bio-Rad, Hercules, CA, USA), according to the manufacturer’s protocol. Briefly, 20 ng of the total RNA was reverse transcribed into complementary DNA (cDNA) by reverse transcriptase for 30 min at 50 °C. Quantitative polymerase chain reaction (PCR) was carried out using SYBR Green as the indicator (Bio-Rad, Hercules, CA, USA). Each reaction mixture consisted of 22 μL iQ SYBR Supermix, 1 μL of cDNA, 1 μL forward primer and 1 μL reverse primer. The sequences of primers used are listed in Table 1. PCR was performed as follows: pre-denaturation for 2 min at 94 °C, 38 cycles of amplification for 30 s each, at 94 °C, 30 s at 60 °C, and 30 s at 72 °C. This series of cycles were followed by a melt-curve analysis to check the reaction specificity. The values of gene expression levels of Ki67, and MUC5B were normalized against the housekeeping gene, GAPDH for each RNA sample. Values were presented as mean ± standard error of mean (SEM). Student’s *t*-test was used to compare the data between the groups, and *p* < 0.05 was considered statistically significant.

### 4.6. Immunocytochemistry

The immunocytochemical analysis was performed on the HTREC to investigate the distribution and morphology of the respiratory cells in the construct. For this purpose, after the removal of the culture medium, the HTREC construct was fixed using 4% (*w*/*v*) paraformaldehyde (Sigma-Aldrich, St. Louis, MO, USA) for 20 min at 4 °C followed by washing three times with PBS (Invitrogen, Waltham, MS, USA). Cell permeabilization was performed using 0.1% (*v*/*v*) Triton-X (Sigma-Aldrich, St. Louis, MO, USA) with 10 min incubation at room temperature followed by washing three times with PBS. Nonspecific binding was blocked using 10% (*v*/*v*) goat serum (Thermo Fisher, Waltham, MA, USA) at 37 °C for 1 h. Mouse anti-MUC5AC antibody with 1:200 dilution ratio in 1% (*v*/*v*) goat serum and mouse anti-Ki67 primary antibody with 1:200 dilution ratio in 1% (*v*/*v*) goat serum were used as primary antibodies at 4 °C for overnight incubation in the dark. The construct was then washed three times with PBS followed by incubation with secondary antibodies, polyclonal conjugated anti-mouse antibody at 37 °C for 2 h. The cells then were washed three times with PBS and were counterstained with 4′,6-Diamidino-2- Phenylindole, Dihydrochloride (DAPI) (Life Technologies, Carlsbad, CA, USA) diluted in PBS with 1:15,000 dilution ratio at room temperature for 20 min. DAPI counterstaining was performed to understand the cellular distribution and total population. The stained cells were visualized using a confocal microscope (Nikon A1, Tokyo, Japan). Image capturing for each of the HTREC (*n* = 3) was repeated with three predetermined positions (field of view), and a representative image was presented as a result. The same fields of view that were used for image capturing were also used for obtaining the number of cells positive for Ki67 or MUC5AC on day 1 and 4 post-incorporation of RECs into HTREC and the counting was performed manually based on visual detection of the cells under the confocal microscope.

## 5. Conclusions

We have successfully isolated RECs from nasal turbinate with their phenotype resembling the native ones. The scaffold comprising of human blood plasma polymerized with calcium chloride was fabricated, and it was proven that the construct is supportive of the cell proliferation and mucin secretion phenotype of RECs, for at least 4 days post-REC incorporation. The finding proves that HTREC can be a suitable candidate for respiratory epithelial tissue reconstruction.

## Figures and Tables

**Figure 1 molecules-26-06724-f001:**
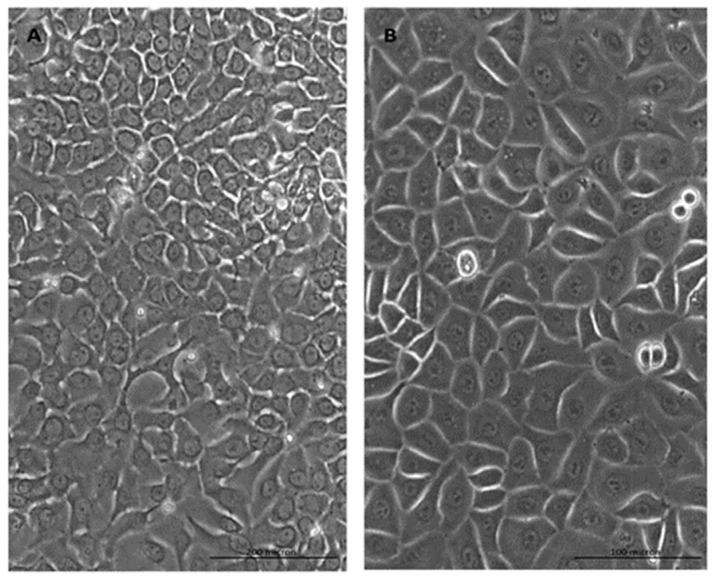
Monolayer human respiratory epithelial cells (RECs) cultured in a 6-well plate. Presented RECs were obtained from a co-culture of RECs and human fibroblasts and at passage 1, the cells showed polygonal morphology. (**A**,**B**) show 100× and 200× magnifications, respectively.

**Figure 2 molecules-26-06724-f002:**
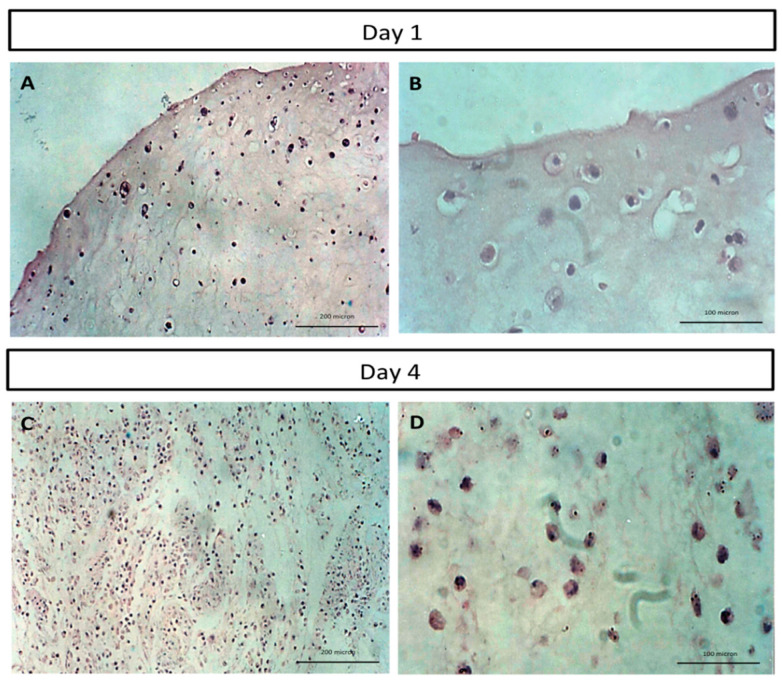
Cross-section view of the hematoxylin and eosin-stained human tissue respiratory epithelial construct (HTREC): (**A**,**B**) show a layer of the construct with respiratory epithelial cells residing within the calcium chloride polymerized-human plasma at day 1 with 100× and 200× magnifications, respectively. (**C**,**D**) show the same construct at day 4 with 100× and 200× magnifications, respectively. The cells were found expanding and proliferating within the HTREC. Results are from a representative of three biologically independent experiments.

**Figure 3 molecules-26-06724-f003:**
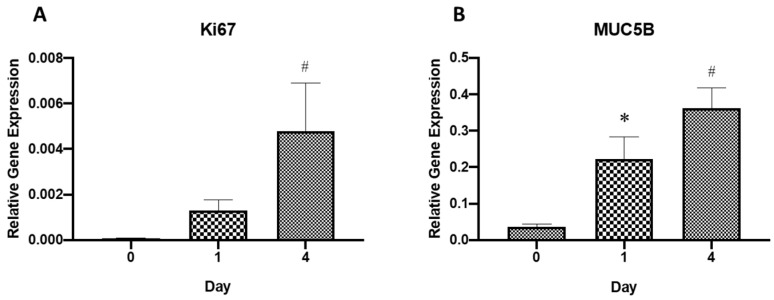
Relative gene expression level of Ki67 (**A**) and MUC5B (**B**) genes in cultured respiratory epithelial cells within the human-tissue-engineered respiratory epithelial construct at day 0, 1 and 4. The gene expression levels were normalized against the housekeeping gene, GAPDH. Values are presented as mean ± standard error of mean. Student’s *t*-test was used to compare the data between the groups, and *p* < 0.05 was considered statistically significant. * *p* < 0.05; compared values of human-tissue-engineered respiratory epithelial construct at day 0 and 1 and # *p* < 0.05; compared values of human-tissue-engineered respiratory epithelial construct at day 0 and 4.

**Figure 4 molecules-26-06724-f004:**
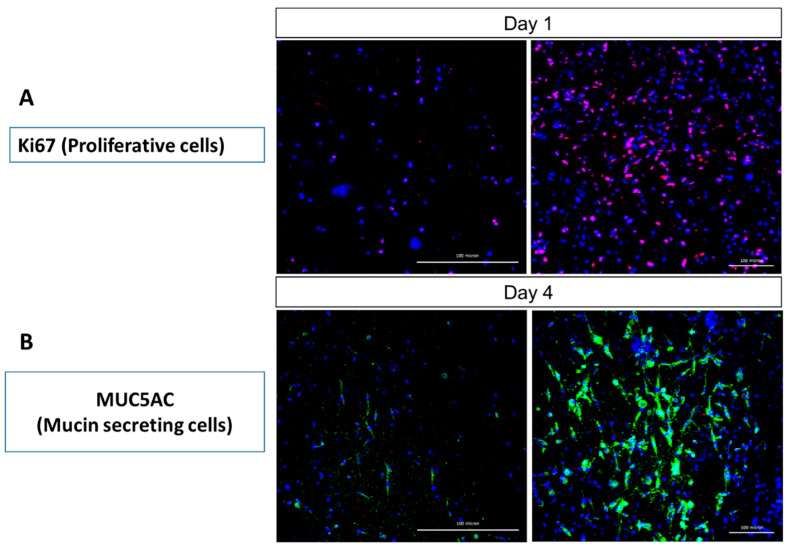
Immunohistochemistry of the human tissue respiratory epithelial construct (HTREC) with residing respiratory epithelial cells (RECs), immunostained with (**A**) anti-Ki67 antibody (red) as the proliferation marker and (**B**) anti-MUC5AC antibody (green) as the mucin secretion marker. The cell nuclei were counter-stained with 4′,6-diamidino-2-phenylindole (DAPI) (blue). The RECs were found to be proliferating (**A**) and secreting mucin (**B**) while residing within the CaCl_2_-polymerized human plasma. The scale bar represents 100 μm. The results are representative of three biologically independent experiments.

**Figure 5 molecules-26-06724-f005:**
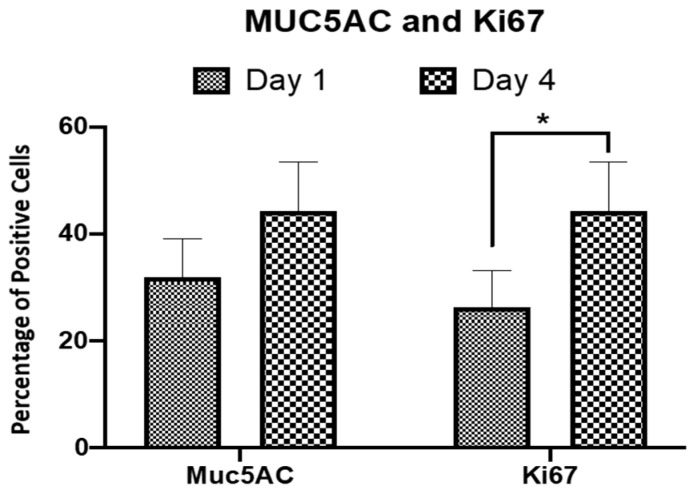
Percentage of mucin secreting and proliferative respiratory epithelial cells residing within the human tissue respiratory epithelial construct. The population of mucin secreting cells increased from day 1 to day 4. Similarly, the percentage of proliferative cells increased from day 1 to 4. Values are presented as mean ± standard error of mean. Two-way ANOVA was used to compare the data between the groups, and * *p* < 0.05 was considered statistically significant.

**Table 1 molecules-26-06724-t001:** Primer sequences used in real-time PCR for quantitative gene expression analysis of respiratory epithelium.

Gene	Accession No.	Primers 5′ → 3′
GADPH	BC0203308	F: 5′-TCC CTG AGC TGA ACG GGA AG-3′ R: 5′-GGA GGA GTG GGT GTC GCT GT-3′
MUC5B	U95031	F: 5′-GTC AAC AGC CAT GTG GAC AAC-3′ R: 5′-CTC CTC ACA GGA GTA GCA GCA-3′
Ki67	NM-002417	F: 5′-GGC TCT AGA GGA CCT GGT TGG-3′ R: 5′-GCT GAC TGC TAG GGG CTC TTC-3′

## Data Availability

Not applicable.

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
