# Peer review of "Evaluating Feasibility of Human Tissue Engineered Respiratory Epithelium Construct as a Potential Model for Tracheal Mucosal Reconstruction"

_molecules, 2021, doi:10.3390/molecules26216724_

Round 1

Reviewer 1 Report

The authors introduced a scaffold comprising of human blood plasma polymerized with calcium chloride for use to culture RECs isolated from nasal turbinate. Based on the findings, they concluded that the construct is supportive to the cell proliferation and mucin secretion phenotype of RECs.

Their idea is of interest but there are some major concerns need to be clarified.

  1. Although the authors used their published method for RECs isolation, I do believe that the trypsinization technique can not completely eliminate the fibroblasts from the culture dishes. The authors should show the evidences of expression of RECs specific markers or absence of fibroblast marker. For example, the cytokeratin expression that they used in their previous published.
  2. About the ability to support cell proliferation of their new contruct

- Why did the authors not show the total cell number increased from day 1 to 4, rather than only the percentage?

- Why did the authors only set up 4 days follow-up for cell proliferation? I think 4 days follow-up is a very short-term, it did not show the steady increase of cell proliferations. I wonder that if the trypsinization method used in isolation step can affect the longterm proliferation of your RECs?

  1. As I understand, the authors looked at MUC5B expression with PCR, and MUC5AC with immunocytochemistry to investigate differentiation of respiratory epithelial cells. Is that right?

- If not, please clarify your purpose.

- If yes, why did you only look at MUC5B expression, instead of other markers which are more specific

Author Response

Dear Reviewers,

Journal

Molecules

Manuscript ID

Molecules-1434065 

Title of Paper

Evaluating Feasibility of Human Tissue Engineered Respiratory Epithelium Construct as a Potential Model for Tracheal Mucosal Reconstruction

Authors

Mohd Heikal Mohd Yunus, Zahra Rashidbenam, Mh Busra Fauzi, Aminuddin Bin Saim, Ruszymah Binti Hj Idrus

Thank you for your letter and the opportunity to revise our paper on “Evaluating Feasibility of Human Tissue Engineered Respiratory Epithelium Construct as a Potential Model for Tracheal Mucosal Reconstruction”. We are grateful to the editors and reviewers for their dedicated time and constructive comments on our manuscript. The suggestions offered by the reviewers have been immensely helpful.

We have addressed all the issues indicated in the review report. Changes in the initial version of the manuscript are either highlighted for added sentences or strikethrough for deleted sentences in the revised version with tracked changes. We look forward to the outcome of your assessment.

We hope the revised manuscript will better suit the Molecules requirement but are happy to consider further revisions, and we thank you for your continued interest in our research.

Yours sincerely,
Corresponding author

Mohd Heikal Mohd Yunus

MD, MMedSci, PhD

Reviewer Comments, Author Responses and Manuscript Changes

Response to Reviewer 1

*** All citations and references are updated after correction.

*** Email address of one of the authors (Zahra Rashidbenam) is corrected (highlighted in     

      yellow).

The authors introduced a scaffold comprising of human blood plasma polymerized with calcium chloride for use to culture RECs isolated from nasal turbinate. Based on the findings, they concluded that the construct is supportive to the cell proliferation and mucin secretion phenotype of RECs.

Their idea is of interest but there are some major concerns need to be clarified.

  1. Although the authors used their published method for RECs isolation, I do believe that the trypsinization technique cannot completely eliminate the fibroblasts from the culture dishes. The authors should show the evidences of expression of RECs specific markers or absence of fibroblast marker. For example, the cytokeratin expression that they used in their previous published.

Thank you for raising this point.

We followed established method for isolating RECs from nasal turbinate. Isolation of RECs using co-culture method from nasal turbinate and via differential tripsinization is routinely practiced in our laboratory. We had extensively characterised the isolated RECs from nasal turbinate in terms of REC specific markers (MUC5B and MUC5AC) in our current study and in our pervious papers. Reference to our previous reports are already cited in Result (2.1 Cell Morphology) and Discussion section as follows:  

Result (2.1 Cell Morphology): “In our previous studies, we successfully characterized the RECs isolated from nasal turbinate via gene expression (CK18 and 14, MUC5AC and Ki67) [36] and immunocytochemical analysis (acetyl β-tubulin, CK14, MUC5AC and Ki67) [36,37].”

Discussion: “The RECs were isolated following an established protocol [10] by which the expression of CK14 and 18, MUC5AC and Ki67 [36] and immunocytochemical expression of markers acetyl β-tubulin, CK14, MUC5AC and Ki67 were proven [36,37].”

In regards to your concern on fibroblast contamination, actually after differential trypsinization at passage 0, in order to reach desired number of RECs, we needed to go through several passaging for cell expansion.  Several passaging removes any possible fibroblast contamination. Moreover, fibroblast cannot proliferate for long time in serum free media (LHC-9 culture medium), which we used, for growth and maintenance of our isolated RECs and this further prevents RECs from being contaminated by fibroblasts.

The respective section is highlighted in yellow within manuscript.

  1. About the ability to support cell proliferation of their new contract

- Why did the authors not show the total cell number increased from day 1 to 4, rather than only the percentage?

Thank you for your raising this question.  To normalise our data, we presented the cell number positive for mucin secretory marker (MUC5AC positive) and proliferation marker (Ki67 positive) as cell percentage (positive cells/total number of cells x100).   We normalise our data to make if feasible for other researchers to reproduce our data, since other researchers might use different initial cell seeding number and different dimension of scaffold (different cell seeding ratio).   

- Why did the authors only set up 4 days follow-up for cell proliferation? I think 4 days follow-up is a very short-term, it did not show the steady increase of cell proliferations. I wonder that if the trypsinization method used in isolation step can affect the longterm proliferation of your RECs?

Thank you for asking this. We suggested using 4 days follow-up for cell proliferation because of the following reasons:

a)      We had restriction in regards to the size of the scaffold we used. We had to make small scaffold in terms of dimension in order to be able to maintain it in a well of 12-well plate and for feasibility of cell incubation. Having that said, exceeding cell incubation period for longer time such as a week or so, would result in cell growth plateau, which is not a favourable condition.

b)      Our data is actually preliminary. To emphasis on this point and on the necessity of further studies on supporting ability of our scaffold for longer period of time, the following section is added into discussion part:

“ In future studies, longer period for evaluating cell proliferation and mucin secretion by residing REC in CaCl2-polymerised human plasma is necessary  ”

The respective section is highlighted in yellow within manuscript.

c)       The polymerised plasma cannot be maintained for very long time and it can degrade easily. Since polymerised plasma can degrade easily, this feature is much favourable in clinical context. In our big animal study (Heikal MYM, et al. Autologous implantation of bilayered tissue-engineered respiratory epithelium for tracheal mucosal regenesis in a sheep model. Cells Tissues Organs 2010; 192: 292–302) the polymerised plasma with its residing cells is often transplanted within two days and in that sense 4 days evaluation of cell proliferation and mucin secretion was suitable for our purpose.  We had also choose blood plasma with its short-term stability in this study because we prefer autologous material in future for clinical trial.

Meanwhile, our results including histology, gene expression analysis (Ki67), Immunocytochemistry (Ki67+) and percentage of proliferating RECs (Ki67+) all in consistence to each other were showing significant increase (for gene expression and cell percentage) in cell proliferation of RECs while resided within CaCl2-polymerised human plasma.  So the trypsinization method we used does not seem to compromise proliferation of isolated RECs.

Regarding your concern on the trypsinization method used in isolation step and whether it can affect the long-term proliferation of RECs:

We followed established method for isolating RECs from nasal turbinate and isolation of RECs using co-culture method from nasal turbinate and via differential tripsinization is routinely practiced in our laboratory. We had extensively characterised the isolated RECs from nasal turbinate in terms of proliferation specifically (Ki67 expression) and it was reported in our pervious papers and it was already cited in Result (2.1 Cell Morphology) and Discussion section as follows:  

Result (2.1 Cell Morphology): “In our previous studies, we successfully characterized the RECs isolated from nasal turbinate via gene expression (CK18 and 14, MUC5AC and Ki67) [36] and immunocytochemical analysis (acetyl β-tubulin, CK14, MUC5AC and Ki67) [36,37].”

Discussion: “The RECs were isolated following an established protocol [10] by which the expression of CK14 and 18, MUC5AC and Ki67 [36] and immunocytochemical expression of markers acetyl β-tubulin, CK14, MUC5AC and Ki67 were proven [36,37].”

The respective section is highlighted in yellow within manuscript.

  1. As I understand, the authors looked at MUC5B expression with PCR, and MUC5AC with immunocytochemistry to investigate differentiation of respiratory epithelial cells. Is that right?

- If not, please clarify your purpose.

- If yes, why did you only look at MUC5B expression, instead of other markers which are more specific

Thank you for pointing this out.  Yes, we looked at MUC5B with PCR and MUC5AC with immunocytochemistry analysis.

We had already discussed about the importance, specificity and significance of mucin secretion and that the MUC5B and MUC5AC are the most readily detected markers in healthy airway epithelial cells. We would like to kindly drag your attention to the following section which was already stated in discussion section:

“In a most recent study on localizing the mucin markers expression on normal/healthy human airway, it had been found that MUC5AC is specifically localized on proximal cartilaginous airway and it is co-expressed with club cells secretory protein [43]. Hence, detection of CK18 and MUC5AC (as a marker of mucin secretory cells) expression in isolated RECs from nasal turbinate confirms the proper and expected phenotype of isolated cells [44]. Among the expressed polymeric secreted mucin markers in the airway, the MUC5AC and MUC5B are the most abundant ones [45] and the significance of maintaining and promoting mucin secretory phenotype by RECs relies of the role they play in first line of defence in innate immune system. Mucin binds to infectious agents; has antioxidant, antiprotease, and antimicrobial properties [46] and the combined function of mucin and cilia clears the airway from variety of pathogens and particles inhaled from the external environment [47].”

We kindly drag dear reviewer’s attention to the following sentences in discussion section

 “Further investigations on suitability of the HTREC in supporting the cilia formation and expression of CK14 and CK18 (as markers of cell proliferation) by its residing RECs are required”

The respective section is highlighted in yellow within manuscript.

Reviewer 2 Report

The presented work is certainly interesting, but I have the impression that it is still at an early stage of research (little experience, a few days of comparative analyzes) to confirm the assumptions presented. I think the authors should explain what needs to be done to confirm that HTREC may be a suitable candidate for the reconstruction of airway epithelial tissue. Below are my other comments:

line 97-105, the scientific novelty of the conducted research is not emphasized enough, please underline it.

line 125-126, why was plasma polymerized with CaCl2? could another salt meet the same conditions? why CaCl2 is the best? I need explanation please.

line 127-128, why did the studies compare day 1 to day 4? what was it caused by?

line 169-171, which is the dominant mechanism then? What is the evidence that more mucin is secreted?

line 287-290 please discuss what future studies will need to be done to confirm HTREC as a promising material for tracheal restoration?

Author Response

Dear Reviewers,

Journal

Molecules

Manuscript ID

Molecules-1434065 

Title of Paper

Evaluating Feasibility of Human Tissue Engineered Respiratory Epithelium Construct as a Potential Model for Tracheal Mucosal Reconstruction

Authors

Mohd Heikal Mohd Yunus, Zahra Rashidbenam, Mh Busra Fauzi, Aminuddin Bin Saim, Ruszymah Binti Hj Idrus

Thank you for your letter and the opportunity to revise our paper on “Evaluating Feasibility of Human Tissue Engineered Respiratory Epithelium Construct as a Potential Model for Tracheal Mucosal Reconstruction”. We are grateful to the editors and reviewers for their dedicated time and constructive comments on our manuscript. The suggestions offered by the reviewers have been immensely helpful.

We have addressed all the issues indicated in the review report. Changes in the initial version of the manuscript are either highlighted for added sentences or strikethrough for deleted sentences in the revised version with tracked changes. We look forward to the outcome of your assessment.

We hope the revised manuscript will better suit the Molecules requirement but are happy to consider further revisions, and we thank you for your continued interest in our research.

Yours sincerely,
Corresponding author

Mohd Heikal Mohd Yunus

MD, MMedSci, PhD

Reviewer Comments, Author Responses and Manuscript Changes

*** All citations and references are updated after correction.

*** Email address of one of the authors (Zahra Rashidbenam) is corrected (highlighted in     

      yellow).

Response to Reviewer 2

The presented work is certainly interesting, but I have the impression that it is still at an early stage of research (little experience, a few days of comparative analyzes) to confirm the assumptions presented. I think the authors should explain what needs to be done to confirm that HTREC may be a suitable candidate for the reconstruction of airway epithelial tissue. Below are my other comments:

line 97-105, the scientific novelty of the conducted research is not emphasized enough, please underline it.

Thank you for asking about scientific novelty of the conducted research. To answer this comment, the following section is added into the introduction.

“In their study, combination of chondrocytes and epithelial cells isolated from sheep nasal septum on a basement of glycolic acid were used for fabrication of trachea [31]. In our study, we used a simple approach in fabrication of epithelial construct by using only one cell type (respiratory epithelial cells) on a basement of plasma. Both, the cell isolation and the blood plasma collection are non-invasive methods and in future, both can be available from autologous sources and that makes the construct immune compatible with the recipient.” The respective section is highlighted in yellow within manuscript.

line 125-126, why was plasma polymerized with CaCl2? could another salt meet the same conditions? why CaCl2 is the best? I need explanation please.

Thank you for pointing this out. In our study, human plasma was withdrawn from human whole blood in sodium citrate blood collection tubes. Sodium citrate which is an anticoagulant, binds to the calcium ions (Ca2+) in the plasma, therefore inhibits the initiation stage of clot formation. The sufficient amount of Ca2+ was then added as CaCl2 solution into the admixture of the cells-plasma in order to initiate the clot formation process until a layer of cells-fibrin composite formed.

line 127-128, why did the studies compare day 1 to day 4? what was it caused by?

Thank you for asking this. We suggested using 4 days follow-up for cell proliferation because of the following reasons:

a)      We had restriction in regards to the size of the scaffold we used. We had to make small scaffold in terms of dimension in order to be able to maintain it in a well of 12-well plate and for feasibility of cell incubation. Having that said, exceeding cell incubation period for longer time such as a week or so, would result in cell growth plateau, which is not a favourable condition.

b)      Our data is actually preliminary. To emphasis on this point and on the necessity of further studies on supporting ability of our scaffold for longer period of time, the following section is added into discussion part:

“ In future studies, longer period for evaluating cell proliferation and mucin secretion by residing REC in CaCl2-polymerised human plasma is necessary  ”

The respective section is highlighted in yellow within manuscript.

c)       The polymerised plasma cannot be maintained for very long time and it can degrade easily. Since polymerised plasma can degrade easily, this feature is much favourable in clinical context. In our big animal study (Heikal MYM, et al. Autologous implantation of bilayered tissue-engineered respiratory epithelium for tracheal mucosal regenesis in a sheep model. Cells Tissues Organs 2010; 192: 292–302) the polymerised plasma with its residing cells is often transplanted within two days and in that sense 4 days evaluation of cell proliferation and mucin secretion was suitable for our purpose.  We had also choose blood plasma with its short-term stability in this study because we prefer autologous material in future for clinical trial.

line 169-171, which is the dominant mechanism then? What is the evidence that more mucin is secreted?

Thank you for pointing this out. Based on our gene expression data (Figure 3) both Ki67 (as a marker of cell proliferation) and MUC5B (as a marker of mucin secretion) increased significantly from day 1 to day 4. This means that the increase we detected in mucin secretion presented in Figure 4-Day 4 (immunohistochemical analysis) is as a result of the followings:  a) there are more mucin secretion because of the increase in cell proliferation and b) there is incease in MUC5B gene expression level per se which resulted in more mucin secretion by individual RECs. Having that said, both mechanisms have contribution in increment of mucin secretion at day 4 as compared to day 1 by RECs resided within CaCl2-polymerised human plasma.

The intensity of detected colours in immunohistochemical images represents the level of marker expression. The green colour in Figre 4 represent MUC5AC marker (mucin secretion marker). The intensity of green color is more at day 4 as compared to day 1 and that shows higher expression of MUC5AC –mucin secretory marker at day 4 as compared to day 1.

To elaborate the point mentioned by dear reviewer the following sentences are added into the discussion section.

The respective section is highlighted in yellow within manuscript.

line 287-290 please discuss what future studies will need to be done to confirm HTREC as a promising material for tracheal restoration?

Thank you for this request.  The following section was added into discussion part:

“In future studies, longer period for evaluating cell proliferation and mucin secretion by residing RECs in CaCl2-polymerised human plasma is necessary.  Further investigations on suitability of the HTREC in supporting the cilia formation and expression of CK14 and CK18 (as markers of cell proliferation) by its residing RECs are required. Use of growth factors such as plant sources to enhance cell proliferation of RECs is also suggested for future explorations [60]”.

The respective section is highlighted in yellow within manuscript.

Round 2

Reviewer 1 Report

The authors have responded to the raised questions adequately.

Reviewer 2 Report

I accept the authors' responses to my remarks and comments. I must emphasize that the authors have provided exhaustive answers. Corresponding corrections have been made to the revised version of the manuscript. I believe that the presented manuscript should be qualified for further stages of evaluation.

Round 3

Reviewer 1 Report

The authors have responded well to the raised questions.